# The Role of Carbon Nanotube Pretreatments in the Adsorption of Benzoic Acid

**DOI:** 10.3390/ma14092118

**Published:** 2021-04-22

**Authors:** Pierantonio De Luca, Carlo Siciliano, Anastasia Macario, Jànos B. Nagy

**Affiliations:** 1Department of Mechanical, Energy and Management Engineering, University of Calabria, I-87036 Arcavacata di Rende, Italy; janos.bnagy1@gmail.com; 2Department of Pharmacy, Health and Nutritional Sciences, University of Calabria, I-87036 Arcavacata di Rende, Italy; carlo.siciliano@unical.it; 3Department of Environmental Engineering, University of Calabria, I-87036 Arcavacata di Rende, Italy; anastasia.macario@unical.it

**Keywords:** benzoic acid, carbon nanotubes, functionalization, active carbons, adsorption

## Abstract

Four different types of multi-walled carbon nanotubes (MWCNTs) were used and compared for the treatment of benzoic acid contaminated water. The types of nanotubes used were: (1) non-purified (CNTs^UP^), as made; (2) purified (CNTs^P^), not containing the catalyst; (3) oxidized (CNTs^OX^), characterized by the presence of groups such as, –COOH; (4) calcined (CNTs^900^), with elimination of interactions between nanotubes. In addition, activated carbon was also used to allow for later comparison. The adsorption tests were conducted on an aqueous solution of benzoic acid at concentration of 20 mg/L, as a model of carboxylated aromatic compounds. After the adsorption tests, the residual benzoic acid concentrations were measured by UV-visible spectrometry, while the carbon nanotubes were characterized by TG and DTA thermal analyses and electron microscopy (SEM). The results show that the type of nanotubes thermally treated at 900 °C has the best performances in terms of adsorption rate and amounts of collected acid, even if compared with the performance of activated carbons.

## 1. Introduction

Carbon nanotubes are materials that are increasingly known and studied thanks to their peculiar properties and their high versatility that make them effective in many sectors. They can be described as coiled sheets of graphene that form nanometer-sized tubular structures [1,2,3,4]. Carbon nanotubes can be advantageously used above all in the treatment of water contaminated by pollutants, thanks to their extraordinary adsorbing properties [5,6,7,8,9]. They often exhibit competitive adsorbing properties with the most common adsorbing materials, such as natural materials [10,11], chitosans [12,13], activated carbon [14,15] and microporous materials [16,17]. Lately, many studies report the advantageous use of carbon nanotubes in the treatment of water contaminated by organic pollutants, such as hydrocarbons [18,19,20,21], dyes [22,23,24,25,26] and heavy metals [27,28,29].

In literature, many studies are reported on the removal of benzoic acid both because it can be considered as representative of aromatic organic pollutants and because it is a potentially dangerous pollutant. Benzoic acid is an important chemical preservative used mainly in the food industry. Although many countries have banned and limited its use as a food additive, it is often present in industrial wastewater, capable of causing risks to the environment and human health [30]. The main sources of benzoic acid in the environment are coal refining plants, paper production plants, and the agricultural sector [31]. Many studies have highlighted the effects of benzoic acid and its derivatives on the quality of the water and on the health of its inhabitants. Furthermore, the benzoic acid in the environment can undergo transformations, for example, into halogenated or hydroxylated forms, which are often more dangerous [32,33]. The treatment of waters contaminated by benzoic acid is attracting a lot of interest in researchers who are increasingly active in researching new and more effective methods of removal. In recent years, much research has been aimed at studying the adsorption of benzoic acid with carbon nanotubes. In 2009, Kotel et al. showed that both chemical and thermal pretreatments on carbon nanotubes can significantly improve the ability to adsorb benzoic acid [34]. In 2018, Dai et al. reported a study on the adsorption of benzoic acid through new microporous nano composites synthesized on multi-walled carbon nanotubes, functionalized with different methods. These new nano composites showed excellent adsorbent performances and potential applications for the removal from polluting waters of dangerous organic materials such as benzoic acid [35].

Recently, some researchers have shown that single-walled nanotubes synthesized by the thermal decomposition of ferrocene (Fe(C_2_H_5_)_2_) are able to efficiently remove benzoic acid with higher efficiencies than multi-walled nanotubes. Through simulation studies of molecular dynamics, they have shown that the high efficiency of single-walled carbon nanotubes is attributable to sites with high bonding energy [36].

Other studies report the use of SWNT functionalized with carboxylic groups, where the low degree of oxygenation of the surface of the carbon nanotubes leads to a greater adsorption of benzoic acid [37].

The aim of this work was to determine the best adsorption performance of purified, non-purified, oxidized, and calcined MWCNTs. The results are compared with the adsorption properties of activated carbon, used as a reference. Specifically, different types of carbon nanotubes were synthesized, with different chemical–physical characteristics, and tested for the removal of benzoic acid in aqueous solutions.

The post-process analyses were performed by UV spectrophotometry for the residual concentration in solution and thermogravimetric analysis to confirm the presence of benzoic acid on the walls of the carbon nanotubes. Finally, the different samples after the adsorption tests were observed with scanning electron microscopy (SEM).

## 2. Materials and Methods

### 2.1. Materials

The aqueous solution was prepared using commercial benzoic acid (Carlo Erba, 99.9% weight, Milan, Italy) with a concentration 20 mg/L equal to 0.164 nM, as representative of the waste waters [38]. Activated carbon was purchased (Merck, Darmstadt, Germany). Unpurified, purified and oxidized carbon nanotubes were synthesized as already reported in previous studies [19,39].

Carbon nanotubes calcined at 900 °C were prepared during this research. In particular, the pretreatments of the nanotubes are reported below.

In the previous work [39], the purified carbon nanotubes, precursors of non-purified, oxidized and calcined carbon nanotubes, were subjected to N_2_ adsorption, showing an N_2_ capacity equal to 0.85 nmol/g, BET surface m^2^/g equal to 108.78 m^2^/g, an average pore width equal to 103.70 Å and 4–5 concentric layers. Figure 1 displays the structure of benzoic acid.

#### 2.1.1. Preparation of Unpurified Carbon Nanotubes (CNTs^UP^)

The synthetic method used for the production of CNTs^UP^ was the catalytic chemical vapor deposition (CCVD) method. The supported catalyst used was a Co-Fe on NaY (UOP Y-54 DR PWD) zeolite. The metallic percentage was of 5% wt for each. The catalysts were prepared by impregnation method on the powdered zeolitic phase. The source of carbon was ethylene using N_2_ as carrier gas. The synthesis temperature was 700 °C with a reaction time of 20 min. The carbon deposit (CD) of this reaction was of 1447%. These types of carbon nanotubes contained traces of the catalyst (Figure 2a).

#### 2.1.2. Preparation of Purified Carbon Nanotubes (CNTs^P^)

The purified carbon nanotubes were prepared by treatment of the previously prepared unpurified carbon nanotubes (CNTs^UP^*)* dissolving the zeolitic support in HF (Sigma-Aldrich, 40% wt) and avoiding the presence of metallic oxides with several steps of washing. The total process time was 5 days. This treatment allowed the removal of the catalyst (Figure 2b).

#### 2.1.3. Preparation of Oxidized Carbon Nanotube (CNTs^OX^)

The oxidized carbon nanotubes (CNTs^OX^) were prepared by purified carbon nanotubes (CNTs^P^) using a mixture of HNO_3_ (Sigma-Aldrich, St. Louis, MO, USA, 65% wt) and H_2_SO_4_ (Sigma-Aldrich, 99% wt) with a ratio HNO_3_/ H_2_SO_4_ of 0.6 in for 24 h. The carbon nanotubes thus pretreated contained carboxylic groups on the surface and were catalyst-free (Figure 2c). Many studies reported in the literature have shown that these carbon nanotube oxidation treatments lead to functionalization with carboxylic groups [40].

#### 2.1.4. Preparation of Calcined Nanotubes (CNTs^900^)

To prepare calcined nanotubes, the purified nanotubes were treated at 900 °C for 4 h, using N_2_ only. These types of carbon nanotubes were completely free from traces of catalyst and carboxylic groups (Figure 2b).

### 2.2. Adsorption Tests

The adsorption tests were carried out by introducing 0.05 g of the selected adsorbent material into a flask and to which 100 mL of a benzoic acid solution with concentration 20 mg/L was added. The whole system, at room temperature, was subjected to magnetic stirring. At different pre-selected adsorption times, equal to 10, 30, 90, 200 min, the tests were stopped and the solution separated from the solid phase by filtration. The adsorption time was stopped at 200 min because we wanted to study a feasible and practical time interval for an adsorption treatment. The two phases, solid and liquid, were recovered and subsequently subjected to characterization.

### 2.3. Characterization

The concentrations of benzoic acid in aqueous solutions were detected by spectrophotometer UV (UV-3100PC, Shimadzu, Kyoto, Japan) in the range 300–1100 nm.

The adsorption measurements were repeated three times and the values were averaged. The thermo-analytical measurements were performed on the automatic TG/DTA instrument (Shimadzu-60, Shimadzu, Kyoto, Japan) under air flow (50 cc/min with heating rate of 10 °C min^−1^). The morphology of the products were examined on a scanning electron microscope (FEI, Hillsboro, OR, USA). The sample preparation relied on the classical method. About 10 mg of CNTs was suspended in 3 mL ethanol, and the suspension was then deposited on a carbonated Cu-Rh grid. The pH values were measured by portable pH-meter (Bicasa, Monza Brianza, Italy).

## 3. Results

### 3.1. Characterization of Carbon Nanotubes

Non-purified (CNTs^UP^), purified (CNTs^P^) and oxidized (CNTs^OX^) carbon nanotubes were characterized in previous work, to which reference should be made to investigate their characteristics [39]. In summary, they had the following thermal characteristics, shown in Table 1.

Unpurified nanotubes (CNTs^UP^) show a total weight loss of approx. 92%, suggesting that the material is not completely burned. This means that the catalyst is still present in the sample, with a composition of approx. 8% wt of the total weight. They exhibit a single DTG peak at approx. 600 °C related to weight loss. The exothermic DTA peak at about 612 °C is due to the combustion of carbon nanotubes and is indicative of a good graphitization of the product, while the lack of exothermic peaks at temperatures below 600 °C denotes the absence of amorphous phases (in the range 300–350 °C) or synthesized products with low graphitization quality (range 350–450 °C). Furthermore, the well-defined DTA peak shows a homogeneous graphitization of the sample.

The purified carbon nanotubes (CNTs^P^) show a total weight loss corresponding to 100%, accompanied by a corresponding DTG peak at 600 °C, showing that the purification step is well performed.

They exhibit two exothermic DTA peaks: the first at approx. 606 °C and the second at approx. 620 °C. The presence of an exothermic peak at a higher temperature denotes the presence of a part of nanotubes with a very high graphitization. Therefore, it is possible to hypothesize that the purification process improves the graphitization properties of CNTs^P^ by eliminating the defects introduced during the synthesis.

The weight losses of oxidized carbon nanotubes (CNTs^OX^) are close to 99.4%, indicating that they are highly pure. The DTG peaks are four and linked to four different weight losses and exactly at 48.12 °C (weight loss 8.2%), 470.32 °C (weight loss 6.7%), 611.32 °C (weight loss 56.2%) and a shoulder at 695.73 °C (weight loss 28.3%). The DTA peaks are also four peaks, of which the first endothermic at 56.90 °C can be attributed to the loss of humidity due to an induced hydrophilic character of the nanotubes following the oxidation treatment, which allowed immediate water adsorption before analysis. The weight loss at 468.53 °C could be related to CNTs with lower thermal resistance related to oxidation, while the third exothermic peak, connected with a weight loss at 604.17 °C, is typical of carbon nanotubes. The weight loss at 693.13 °C could be attributed to CNTs with a higher degree of graphitization [39].

Carbon nanotubes calcined at 900 °C (CNTs^900^), on the other hand, were characterized in this work by TG, DTA and DTG thermal analysis (Table 1).

The TG curve shows, as expected, that the weight loss is 100%, this is because the calcined carbon nanotubes were prepared starting from the purified nanotubes, and therefore traces of the catalyst were no longer present. The DTG curve shows two peaks, the first not very evident below 100 °C. The second DTG peak is at 637.11 °C

From the DTA thermal curves, it can be seen that the CNTs^900^ have an exothermic DTA peak around 628.92 °C. This means that the nanotubes obtained by calcination at 900 °C have a higher degree of graphitization than CNTS^UP^ which have a lower DTA peak. Furthermore, compared with oxidized nanotubes, they have a single peak at a temperature above 600 °C, which denotes a more homogeneous graphitization, although the peak does not reach the higher temperature of 693.13 °C, which can be attributed to a fraction of nanotubes of highly graphitized CNTs^OX^ carbon.

### 3.2. Adsorption of Benzoic Acid on CNTs

From the data shown in Figure 3 it is possible to highlight that the four different types of nanotubes, all allow a lowering of the concentration of benzoic acid, albeit in a different way, confirming a general high adsorbing capacity toward benzoic acid.

Purified nanotubes (CNTs^P^) seem to be the worse adsorption agents. In fact, the amount of solute is reduced up to 50% of total concentration after 1.5 h of adsorption treatment.

Unpurified CNTs^UP^ are better adsorption agents if compared with the CNTs^P^. This type of CNTs can decrease the acidic concentration up to 50% after 0.5 h, reaching a stable value after of treatment.

This performance improvement may be due to the presence of the catalyst bonded to the carbon nanotubes which, as reported, consists of zeolitic material which also has an adsorbent character. It has also been noted that non-purified carbon nanotubes have a higher dispersion in water than purified ones, probably due to the presence of the zeolite material, which is hydrophilic. Additionally, they were in the presence of a catalyst that physically reduces the formation of agglomerates of nanotubes.

This greater dispersing capacity could positively influence the final adsorbing capacity of the non-purified carbon nanotubes.

Oxidized carbon nanotubes are highly efficient in removing benzoic acid, in fact after 10 min the concentration of benzoic acid is reduced by about 50%. This high adsorbing capacity is due to the presence of carboxylated groups generated by the oxidation treatment, which interact with the benzoic acid molecules.

CNTs^900^ show to have the highest adsorption capacity. The calcination treatment promotes the breakdown of interactions between nanotubes, which cause the formation of aggregates of nanotubes, with the consequence of having a better dispersion in the adsorption system.

In fact, the color of the acidic solution after introduction of CNTs^900^ is completely black, while it is possible to note black aggregates during the adsorption treatment with other kinds of nanotubes.

Considering a 30 min average time, the efficiency scale is: CNTs^900^ > CNTs^OX^ > CNTS^UP^ > CNT^P^, for which the concentrations of benzoic acid after the adsorption tests are, respectively, 3.06, 7.81, 9.58 and 15.53 mg/L.

Adsorption of benzoic acid was carried out under the same previous conditions, but with activated carbon in order to be able to draw a comparison with the different adsorbent materials used. The results obtained show that CNTs^900^ exhibit better adsorbing properties than activated carbon. The pH values of the solution tend to increase as the contact times increase, in perfect consistency with the reduction in acid concentration.

The following Figure 4 shows the percentage by weight of benzoic acid adsorbed for the different types of carbon nanotubes as a function of the contact time.

The data reported show that carbon nanotubes calcined at 900 °C reach a reduction of about 80% already in the first ten minutes. For the other types of carbon nanotubes, the highest abatement% is reached at a time of 90 min, with values around 72%, 56%, 58%, respectively, for CNTs^OX^, CNTs^UP^ and CNTs^P^.

### 3.3. Adsorption Capacity

The adsorption capacity was studied using the following Equation:(1)$$q_{e}=\frac{m_{adsorbed}}{m_{adsorbent}}=\frac{\left(C_{i}-C_{f}\right)\cdot V}{m_{CNTs}}$$
where *C_i_* and *C_f_* are the initial and final concentration (mg/L) of benzoic acid, respectively; *V* is the volume of treated solution (L), $m_{adsorbed}$ represents the mass of benzoic acid adsorbed and $m_{adsorbent}$ represents the mass of carbon nanotubes. The following Figure 5 reports the adsorption capacity as a function of contact time for each type of carbon nanotube.

The data reported show that the maximum adsorption capacity for all four different types of carbon nanotubes is reached at 90 min of contact time. The adsorption capacity for CNTs^900^, CNTs^OX^, CNTs^P^, CNTs^UP^ at 90 min are, respectively 39.9, 28.9, 23.3, 22.3 mg/g.

The adsorption capacity of activated carbon at 90 min is equal to 30.5 mg/g, a value lower than that of CNTs^900^. All of this allows us to define the CNTs^900^ as the best materials among those analyzed for the adsorption of benzoic acid.

### 3.4. Kinetic Study

The adsorption rates, in the different time intervals, were calculated for each type of carbon nanotube and for the activated carbons, during the adsorption tests (Figure 6). The adsorption rates were calculated using the following expression:Adsorption rate = (C_2_-C_1_)/(t_2_ − t_1_)(2)
where C_1_ and C_2_ represent the concentration of benzoic acid expressed in mg/L at the contact time t_1_ and t_2_, respectively.

The time intervals considered were (0′–10′); (10′–30′); (30′–90′); (90′–200′).

The results obtained show that CNTs^UP^ and CNTs^P^ reach a maximum adsorption value in the time interval (10′–30′) and with fairly low maximum adsorption rates, respectively equal to 0.42 and 0.16 mg/L·min. The greater efficiency of CNTs^OX^ and CNTs^900^ is highlighted by shorter times, less than 10 min, in which the maximum adsorption rate values are reached. The maximum values of the adsorption rate, also comparing them with those relating to activated carbons, are 0.9, 1.2, 1.6 mg/L·min for CNTs^OX^, activated carbon and CNTs^900^, respectively. The set of results show that CNTs^900^ are the best adsorbent materials among those studied, obtaining higher adsorption rate value and shorter adsorption time.

Figure 7 shows the residual concentration of benzoic acid on a logarithmic scale as a function of the contact time for CNTs^UP^, CNTs^P^ and CNTs^OX^.

The adsorption of benzoic acid on CNTs^UP^, CNTs^P^ and CNTs^OX^ are first order because the adsorption is not very strong. The kinetic constants are k_1_ = −0.0034 (min^−1^); −0.0032 (min^−1^) and −0.0029 (min^−1^), R^2^ = 0.4707, 0.5553, 0.2593 for CNTs^UP^, CNTs^P^ and CNTs^OX^, respectively (Figure 7a–c). On the other hand, because the adsorption of benzoic acid is very strong on CNT^900^ and on activated carbon, the reaction is of zero order. The kinetic constant relative to the first linear segment is K_0_ = −1.626 (mg·L^−1^·min^−1^) and −1.235 (mg·L^−1^·min^−1^) for CNTs^900^ and activated carbon, respectively.

### 3.5. Thermal Characterization Post-Adsorption of Solid Materials

Figure 8 shows the TG thermal curves of the different types of carbon nanotubes after the adsorption tests.

From the thermogravimetric curves (TG) it is possible to note two main weight losses: the first occurring in the temperature range between 100 °C and 550 °C and a second between 550 °C and 800 °C.

The first loss is certainly attributable to the presence of benzoic acid, as it has a melting temperature of 121–123 °C and a boiling temperature of 249 °C. Furthermore, these losses cannot be attributed to the carbon nanotubes, as it is known that they are thermally stable up to 600 °C [39]. These losses are of a different entity in the different types of carbon nanotubes used.

The first weight losses, between 100–550 °C, are very low as regards the CNTs^UP^ and CNTs^P^, with values of 10.0% and 12.3%, respectively. For the CNTs^OX^ and CNTs^900^ samples, these are much higher with values of 32.6% and 85%, respectively. These results are perfectly consistent with the abatement data, reported in the previous Section 3.1, which confirms this order of effectiveness in the removal of benzoic acid: CNTs^900^ > CNTs^OX^ > CNTs^UP^ > CNTs^P^. The total weight losses are 100% except for CNTs^UP^, which has a residual weight attributable to the presence of the catalyst, which continues to be present in this type of nanotube, having not undergone any purification treatment.

The activated carbon shows a first weight loss of 25% medium value between the CNTs^UP^, the CNTs^P^, and the CNTs^OX^ and the CNTs^900^, respectively.

The following Figure 9 reports the thermal curves DTg and DTA of the carbon nanotubes after the adsorption tests.

The DTA curves show endothermic peaks at temperatures below 200 °C, which are very evident for CNTs^OX^ and CNTs^900^, as they adsorbed a greater quantity of benzoic acid. These endothermic DTA peaks, accompanied by corresponding DTG peaks, can be attributed to the melting and loss of benzoic acid. In the case of CNTs^OX^ there is also a second endothermic DTA peak, accompanied by a corresponding DTG peak, just over 200 °C, which can be attributed to the interactions of benzoic acid with carboxylic groups present on the functionalized surface of the nanotubes. At temperatures above 600 °C, an exothermic DTA peak is evidently present in all the samples, accompanied by a corresponding DTG peak, which is attributable to the combustion of carbon nanotubes.

### 3.6. SEM Characterization

The observation by the electron microscope (SEM) of carbon nanotubes after the adsorption tests offered useful information to understand the adsorption mechanism of benzoic acid.

Unpurified nanotubes (CNTs^UP^) appear massed, not sufficiently separated and with white spots on the surface attributable to benzoic acid Figure 10a.

The purified nanotubes (CNT^P^) have the classic organization in bundles with dimensions between 10–50 nm. Small white spots denote the presence of benzoic acid on the surface but in less quantity than in non-purified carbon nanotubes Figure 10b.

Oxidized nanotubes (CNTs^OX^) also have small white spots on the surface attributable to benzoic acid. In this case, the presence of functional groups, generated by the oxidation treatment, of the -COOH type guarantee an improvement in the acid adsorption process on the surface of the nanotubes Figure 10c.

The carbon nanotubes calcined at 900 °C have a certainly greater quantity of benzoic acid on the surface than in the other previous cases, which is consistent with the adsorption data reported in Section 3.1, in which the calcined carbon nanotubes are the most efficient of all Figure 10d.

In light of the data obtained, it is possible to state that the different adsorption capacities of the carbon nanotubes used are due to the establishment of distinct phenomena.

In general, adsorption can take place inside the carbon nanotubes or on their surface.

The adsorption on the surface is favored the freer the surface and the greater the separation between the nanotubes. Furthermore, the presence of functional groups on the surface of the nanotubes, such as carboxyl groups, increases their surface adsorption. Even in the absence of functional groups, the adsorption of benzoic acid on the surface is guaranteed by molecular interaction.

Adsorption within the carbon nanotubes is favored by the accessibility of the channels, which depends on a good dispersion of the nanotubes, determined by the absence of agglomerates. Another determining factor is the presence of clean channels, which must be free of amorphous material.

Each of the different treatments carried out on the carbon nanotubes (purification, oxidation, calcination) specifically favors the different adsorption methods, favoring some of them more than others.

In particular, in the purification process, there is the destruction of the zeolite material used as the base of the catalyst. The latter also has an adsorbing capacity; therefore, the better adsorption efficiency of non-purified nanotubes compared with purified ones can be justified by the concomitant presence of microporous adsorbing material.

The oxidation of the nanotubes promotes the superficial adsorption of benzoic acid thanks to the formation of carboxylic groups that promote the reaction with benzoic acid. In addition, it also has a partial cleaning effect of the channels, also favoring an improvement in adsorption within the channels.

Calcination at 900 °C has the predominant effect of destroying interactions between the different nanotubes and therefore of making the carbon nanotubes free of clusters and agglomerates, making them optimally dispersible in water. All this means that the calcination at 900 °C, under nitrogen, allows the carbon nanotubes to play their role fully and optimally as adsorbent materials for the removal of benzoic acid.

The importance of the interaction between the nanotubes was highlighted in an excellent paper by Wisniewski et al. [41]. They have shown that the adsorption of benzene greatly depends on the diameter of the nanotubes and the number of concentric layers. This is due to the formation of interstitial channels within the bundles. However, the individual nanotubes represent the best adsorbing surface, as it is also shown in the present paper.

## 4. Conclusions

In general, all the different types of carbon nanotubes—CNTs^UP^, CNTs^P^, CNTs^OX^, CNTs^900^—showed the ability to adsorb benzoic acid from aqueous solutions, although with different intensities.

The purification treatment leads to a slight lowering of efficiency compared with non-purified carbon nanotubes. This is to be attributed to the fact that the purification, in addition to destroying the catalyst, which is also an adsorbent material and which hinders agglomeration due to its presence, makes the nanotubes less dispersed.

The oxidation and calcination pretreatments at 900 °C proved to be decisive for increasing the adsorption capacity. The oxidation process, which leads to the formation of carboxylic groups, adds a further possibility for the benzoic acid molecules to interact on the surface of the nanotubes, improving the adsorbent performance compared with purified and non-purified nanotubes.

The 900 °C pretreatment of carbon nanotubes was shown to be the best, compared with all types of nanotubes used in this research. In fact, this treatment leads to an important elimination of clusters and therefore to the breakdown of the interactions between the nanotubes, significantly increasing the adsorption surface. An evident effect of this treatment is a complete and uniform dispersion and absence of agglomerates of the CNTs^900^ when they are inserted in the benzoic acid solution.

The CNTs^900^ reach a reduction of benzoic acid of about 80% already in the first ten minutes. For the other types of carbon nanotubes, the highest percentage abatement is reached at a time of 90 min, with values of 72%, 56%, 58% for CNTs^OX^, CNTs^UP^ and CNTs^P^, respectively. Moreover, CNTs^900^ performed better than activated carbon.

Thanks to the results obtained, it is possible to say that the efficiency scale of the different types of nanotubes with respect to the adsorption of benzoic acid from aqueous solutions follows the order: CNTs^900^ > CNTs^OX^ > CNTS^UP^ > CNT^P^.

## Figures and Tables

**Figure 1 materials-14-02118-f001:**
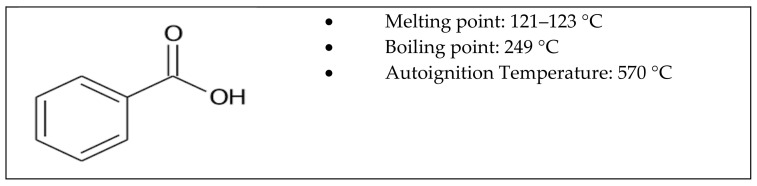
Structure and thermal characteristics of benzoic acid.

**Figure 2 materials-14-02118-f002:**
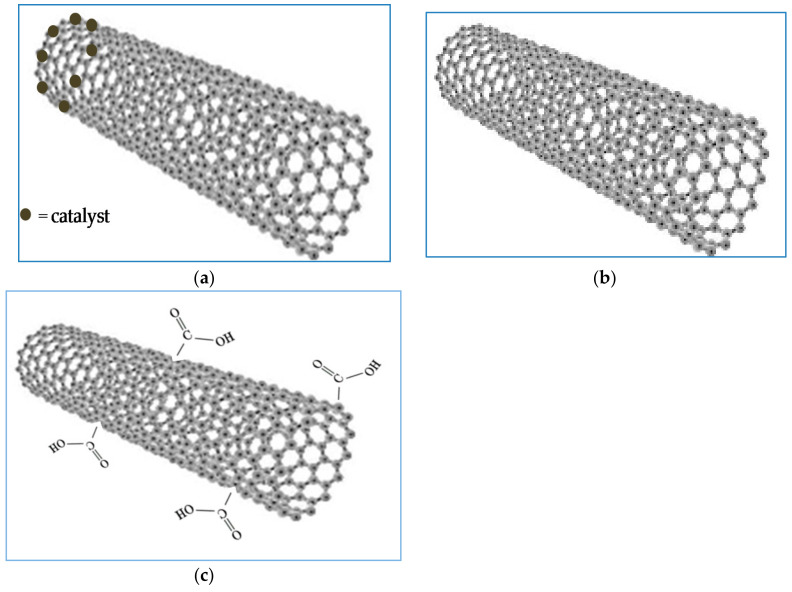
Schematic representation of the main characteristics of the different types of nanotubes used: (**a**) with the presence of catalyst; (**b**) without catalyst and functional groups; (**c**) without the presence of catalyst and with carboxyl groups.

**Figure 3 materials-14-02118-f003:**
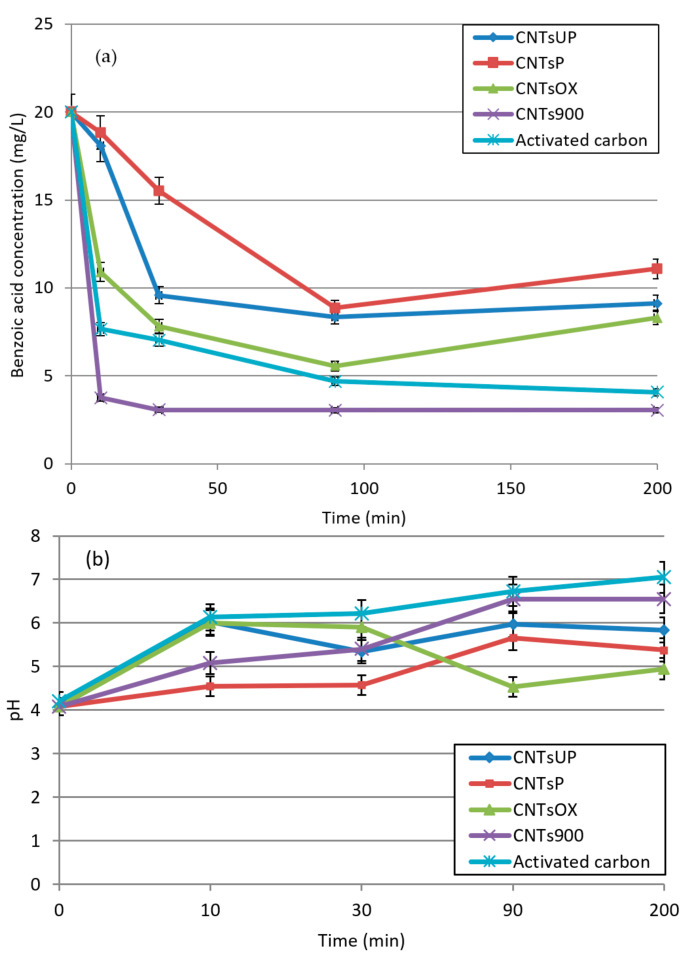
(**a**) Concentration of the benzoic acid and (**b**) pH after adsorption process as a function of time for unpurified (CNTs^UP^); purified (CNTs^P^); oxidized (CNTs^OX^); calcined (CNTs^900^) carbon nanotubes and activated carbon.

**Figure 4 materials-14-02118-f004:**
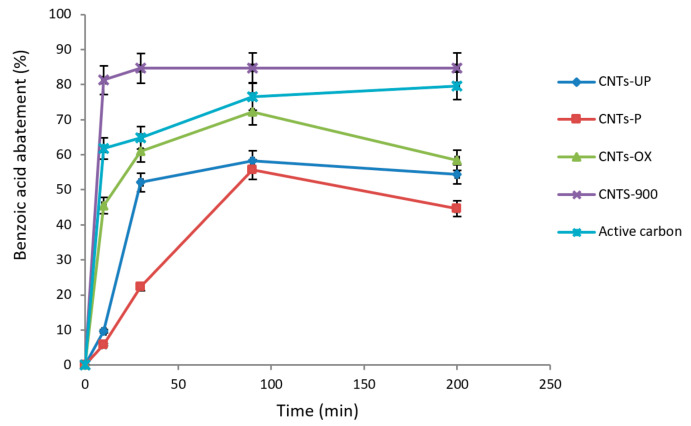
Benzoic acid abatement (%) as a function of contact time for different adsorbent materials.

**Figure 5 materials-14-02118-f005:**
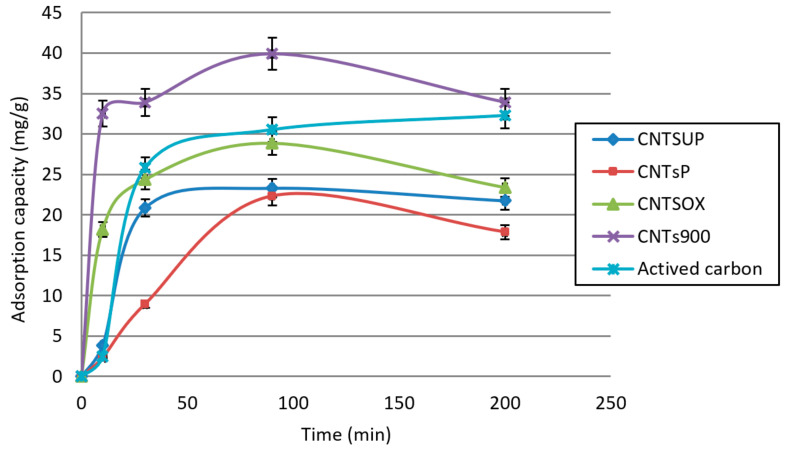
Adsorption capacity as a function of contact time for different carbon nanotube types.

**Figure 6 materials-14-02118-f006:**
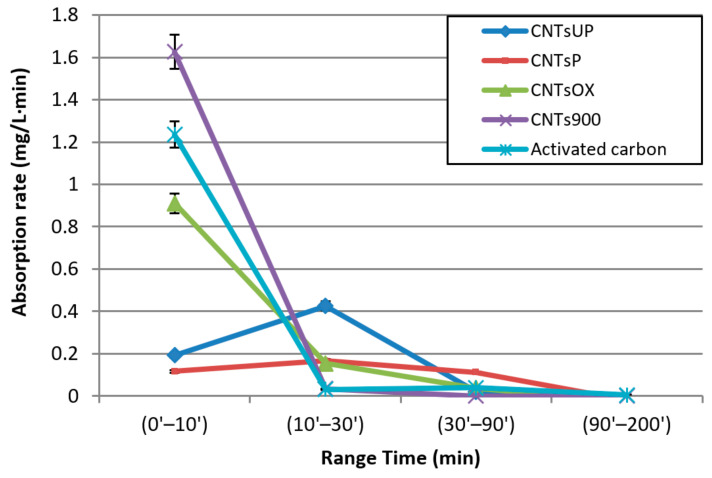
Adsorption rate for different adsorbent materials as a function of range time.

**Figure 7 materials-14-02118-f007:**
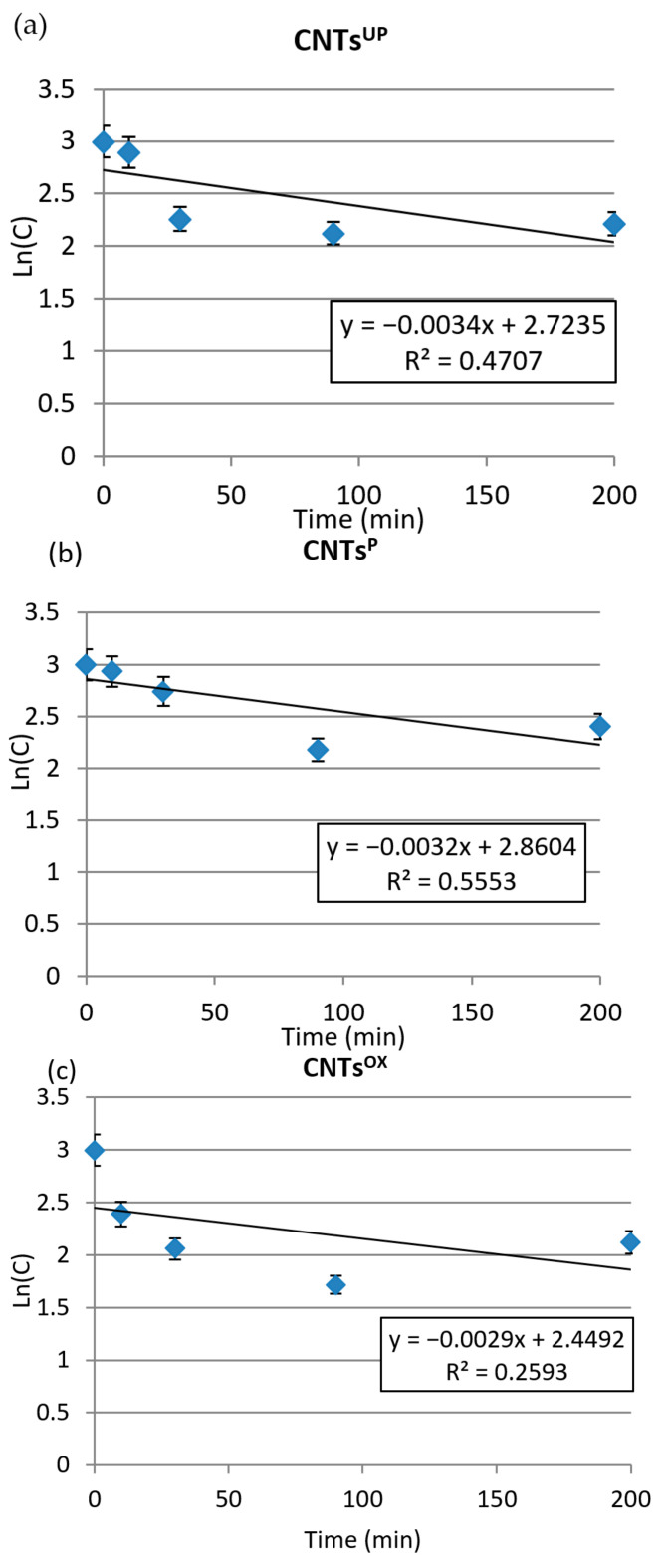
(**a**–**c**) Residual concentrations in logarithmic scale for CNTs^UP^, CNTs^P^, CNTs^OX^ as a function of the contact time.

**Figure 8 materials-14-02118-f008:**
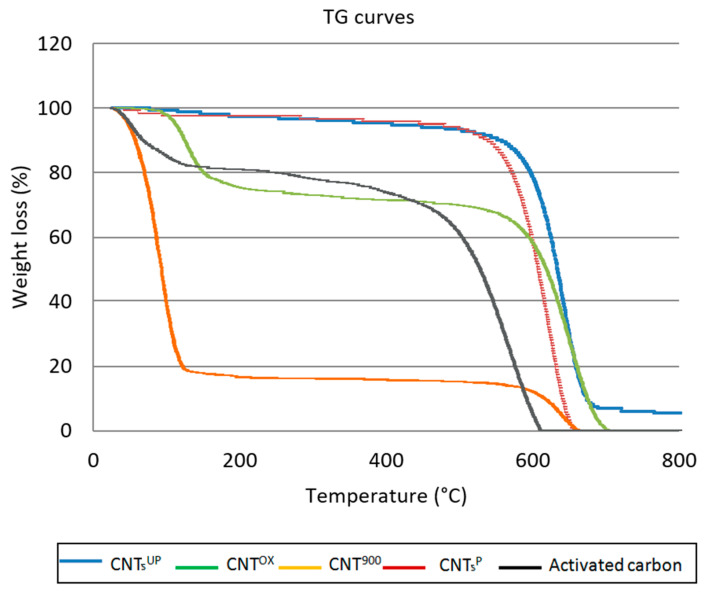
TG curves of the different types of carbon nanotubes and activated carbon after adsorption tests at a contact time of 90 min.

**Figure 9 materials-14-02118-f009:**
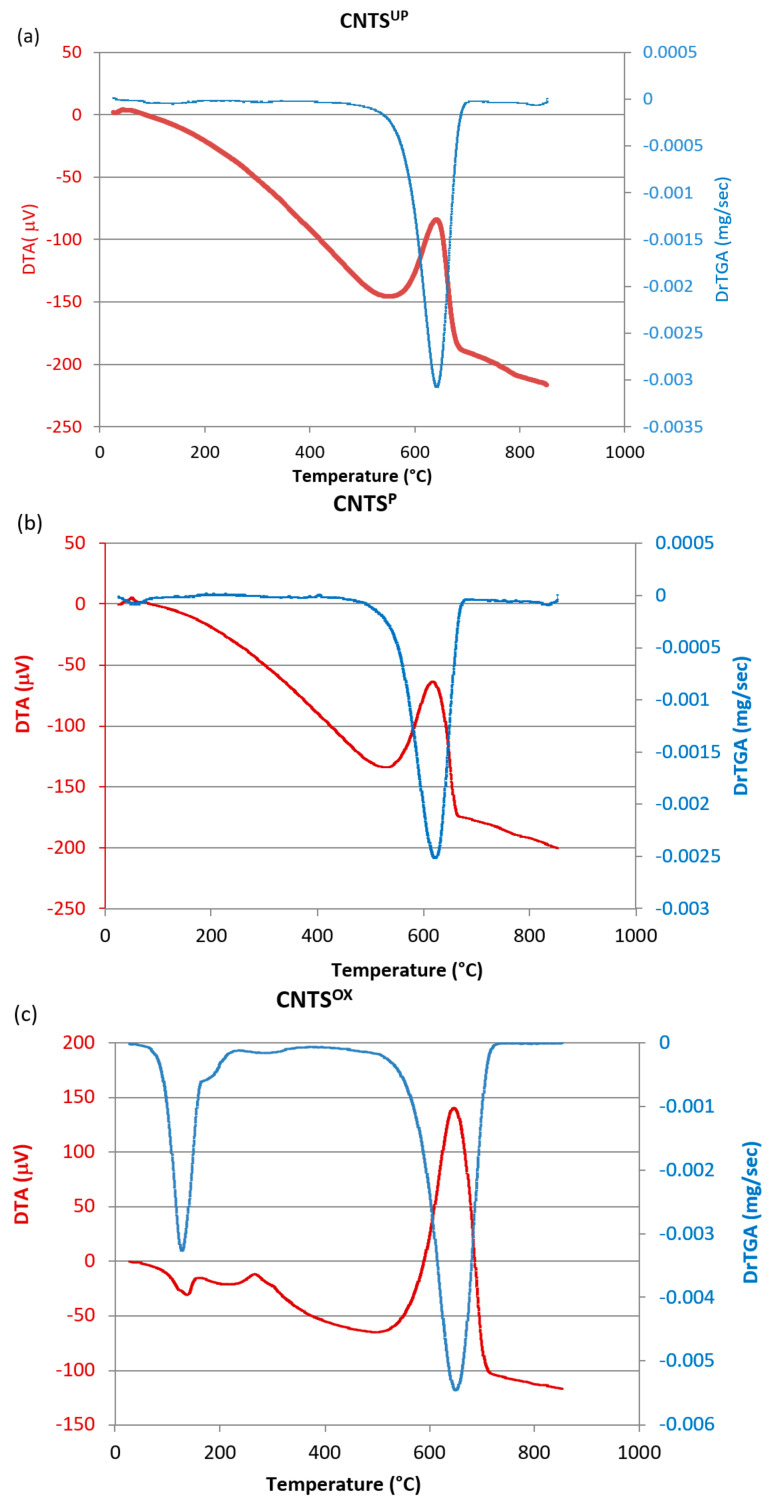
DTG and DTA curves of (**a**) purified; (**b**) unpurified; (**c**) oxidized; (**d**) calcined carbon nanotubes after benzoic acid adsorption tests for a contact time of 90 min.

**Figure 10 materials-14-02118-f010:**
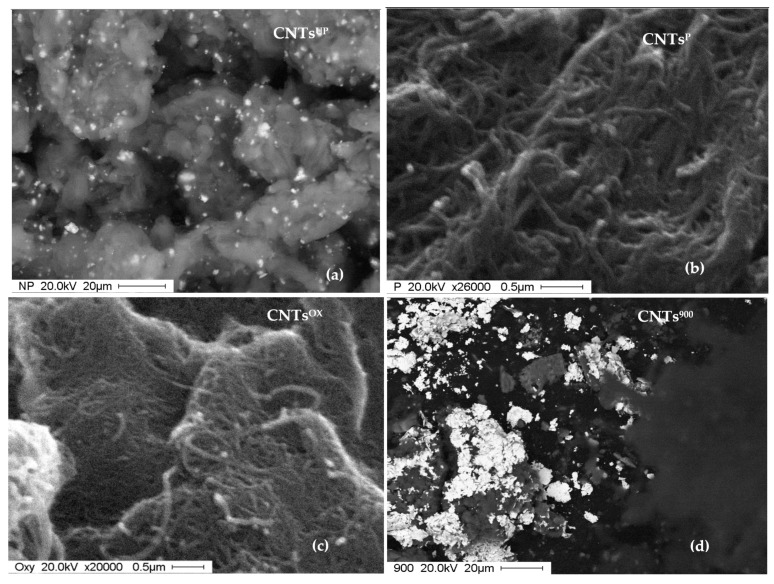
SEM images of the (**a**) purified; (**b**) unpurified; (**c**) oxidized; (**d**) calcined nanotubes after the adsorption test at a contact time of 90 min.

**Table 1 materials-14-02118-t001:** Thermal characteristics of non-purified (CNTs^UP^), purified (CNTS^P^), oxidized (CNTs^OX^) nanotubes and calcined (CNTs^900^).

Carbon Nanotubes	Peaks DTG (°C)	PeaksDTA (°C)	Total Weight Loss (%)
* CNTs^UP^	600 (I)	612 (exo)	92
* CNTs^P^	600 (I)	606 (exo) 620 (exo)	100
* CNTs^OX^	48.12 (I) 470.32 (II) 611.32 (III) 695.73 (IV)	56.90 (endo) 468.53 (exo) 604.17 (exo) 693.13 (exo)	99.4
CNTs^900^	48.4 (I) 637.11 (II)	628.92 (exo)	100

*: Ref. [39].

## Data Availability

Data are contained within the article.

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
