# Peer review of "The Role of Carbon Nanotube Pretreatments in the Adsorption of Benzoic Acid"

_materials, 2021, doi:10.3390/ma14092118_

Round 1

Reviewer 1 Report

The paper presents an attempt to describe the adsorption mechanism of benzoic acid on/at carbon nanotubes and the role of carbon nanotube pretreatments in this process. In my opinion the presentation of the investigation methods as well as the scientific results are not satisfactory for the paper to be recommended for publication. The manuscript should be rejected, but I did give the authors a chance to improve their paper. The minor and major drawbacks to be addressed can be specified as follows:
1.    Introduction. (a) Too many literature references, i.e. 108 out of all 109!!! However, off topic (b) Another problem is the high amount of self-citations in the Introduction, i.e. 18 references [4, 8, 59, 62, 64, 65, 66, 68, 71, 72, 76, 79, 85, 89, 91 96, 98, 99]. (c) Too many paragraphs. (d) Too little space has been devoted to the most important issue, i.e. literature reports on adsorption of benzoic acid on CNTs – lines 64 – 74, only three references (i.e. [107, 108, 109] out of 108!!! Literature on this subject is very wide, see https://scholar.google.pl/scholar?q=%22benzoic+acid%22+adsorption+%22carbon+nanotubes%22&hl=pl&as_sdt=0&as_vis=1&oi=scholart I would prefer 108 papers dealing with this important topic!!!
2.    Page 2, 2.1. Materials. Please provide basic information about the tested materials, the inner (and outer) average diameter and the average number of walls.
3.    Pages 2-4, 2. Materials and Methods. In the case of the paragraphs 2.1.1-4 it is difficult to state which results are new and which are from [89, 91] – see lines 90 – 92. Please make it clear which results were previously published.
4.    Page 3. How do the authors know that carboxylic groups is present there? XPS? FTIR? The Boehm method? Are such results included in [89, 91]? If so, please provide references explain it in the text.
5.    Pages 4 and 5, Tab. 1. Please add data for CNTS900 in Tab. 1.
6.    Page 6, Fig. 3. It would be easier for the reader and very interesting to analyze the results, ((i) weight loss vs. T, (ii) DTA vs. T, and (iii) DTG vs. T) collecting on three panels, i.e. , a comparison for all materials from Tab. 1 and CNTS900 (data taken from Fig. 3). Different points/lines – different colors – see Fig. 5.
7.    Page 7. How many times were the adsorption measurements repeated? 
8.    Page 8, Fig. 4. The results are not very transparent. It would be better to prepare two panels that compare/collect all tested samples. Different points/lines – different colors – see Fig. 5.
9.    Page 9, lines 265 and 266. Too high accuracy of 72.2%, 55.77%, and 58.25%.
10.    Page 10, Fig. 6. (i) Why did the measurements end at 200 minutes? After all, the stabilization has not been achieved. (ii) one panel – five curves.
11.    Page 11, line 296. Reaction rate ---> Adsorption rate. See Fig. 7, y-axis.
12.    Pages 11 and 12, Fig. 7. One panel – five curves.
13.    Pages 12 and 13, Fig. 8. (i) The values of the kinetic constants (i.e. k1 =-0.0034 [min-1], -0.0032[min-1] 319, and -0.0029[min-1]) are very similar. What about errors and statistical analysis? (ii) Is the description of the data satisfactory from the point of view of statistical analysis? (iii) Where are the other two samples?
14.    Pages 13 and 14, Fig. 9. (i) One panel – five curves. (ii) Where is AC sample? See also Fig. 10.
15.    Page 17, Fig. 11. Why are images with different resolutions compared? This is a big mistake. Please repeat the measurements for 0.5 microm.
16.    Page 18, lines 386 – 388. “In the light of the data obtained, it is possible to state that the adsorption of benzoic acid with the different types of carbon nanotubes is promoted by different phenomena that justify the different adsorption capacities of the different types used.” Terrible - 3x different.
17.    Page 18, lines 388 and 389. “In general, adsorption can take place inside the carbon nanotubes or on their surface.” Did the authors check whether the nanotubes opened as a result of the modification? N2 (77K) adsorption?
18.    Page 18, lines 410 and 411. “Calcination at 900°C has the predominant effect of destroying interactions between the different channels.” Are thermogravimetric measurement and microscopy images sufficient to draw such conclusions. The problem is much more complex – see https://doi.org/10.1016/j.jcis.2012.09.026.
19.    Page 20. References [26] and [27] are repeated.
20.    Page 22. “[92] [Gotovac,” ---> “[92] Gotovac,”.

21.    No reference of the results obtained by the authors to previous literature reports during the discussion!!!

Author Response

Dear Reviewer,

Thank you for your time spent reviewing our manuscript and for giving us important tips for improving it. The manuscript has been revised in many parts. The changes have been highlighted in red. We send all the point-by-point answers to your suggestions in the attached files.

We hope that in this form the manuscript will find your approval to be considered for its publication.

We thank you and send you our best regards.

The authors

Reviewer 2 Report

The submitted manuscript describes the quantitative demonstration of the difference in efficiency of benzoic acid removal from water depending on the treatment method of CNTs. The oxidative treatment of CNTsOX showed that the -COOH group established on the surface of CNTs causes interaction with benzoic acid, which has a favorable effect on the adsorption properties. The calcined CNTs900 have a greater adsorption surface thanks to the improved dispersion in water, and the adsorption properties appear to be significantly enhanced. However, I have some parts that I am concerned about. The reason for this and the points to be considered are described briefly as follows.

There are too many line breaks throughout the manuscript, especially in the introduction section, which completely undermines the coherence of the meaning. In other words, every two or three sentences are isolated from each other, and I don't see the essential concept of "paragraphs" that are organized by each story. For example, lines 29-47 should be a single paragraph without line breaks. Please make changes to the paragraph construction for the entire manuscript in the revised version.

I recommend that the authors add a short description of the hazards of benzoic acid with references in the introduction section. If the benefits of removing benzoic acid are clearly described, the readers will find your work to be more valuable.

In lines 88-89, it is stated that a concentration for benzoic acid of 20 mg/L (0.164 nM) is a representative value for wastewater. What is the basis for this? Alternatively, please provide references to support your statement.

The increase in benzoic acid concentration from 90 to 200 minutes for cases other than CNTs900 in Figure 4 is curious. Alternatively, does the concentration at 90 minutes seem to be a little too low? Please add error bars with respect to benzoic acid concentration in Figure 4 to avoid such confusion. It appears from Figure 4 that the benzoic acids are released back into the water after reaching the saturated adsorption amount at 90 minutes. The same can be said for Figures 5 and 6, and I would like to see further discussions in the text, including the addition of error bars to these figures as well. The authors state in lines 288-289 that the maximum adsorption capacity of activated carbon is 30.52 mg/g which is clearly lower than that of CNTs900, but depending on the results of the above discussion, the term "clearly lower" may seem like an overstatement.

In lines 318-319, the authors mention that the adsorption behavior of CNTsUP, CNTsP, and CNTsOX is linear. Can you provide some literature on this subject? I don't see the behavior in Figure 8 as being linear at all, it appears to be 2nd order or higher. Each CNTs may possess a different and complicated kinetics regarding the adsorption behavior of the benzoic acid.

In relation to Figure 9, do you have TG data for activated carbon with benzoic acid adsorbed? Please provide a reference if this has already been shown in previous studies, otherwise I recommend adding the data at 90 minutes to Figure 9 or the electronic supplementary material.

Author Response

(The authors gave the same response as above.)

Round 2

Reviewer 1 Report

The authors revised their manuscript well. The revised version can be accepted for publication.